# A Proof-of-Concept Evaluation of the 1616 Story-Based Positive Youth Development Program

**DOI:** 10.3390/children10050799

**Published:** 2023-04-28

**Authors:** Jean Côté, Jennifer Coletti, Cailie S. McGuire, Karl Erickson, Kelsey Saizew, Alex Maw, Chris Primeau, Meredith Wolff, Brandy Ladd, Luc J. Martin

**Affiliations:** 1School of Kinesiology and Health Studies, Queen’s University, Kingston, ON K7L 3N6, Canada; 2School of Kinesiology and Health Science, York University, Toronto, ON M3J 1P3, Canada; 3Impact Society, Calgary, ON T2E 7L8, Canada; 4Ladd Foundation, Toronto, ON M3J 1P3, Canada

**Keywords:** knowledge to action, pilot test, character development, partnership research

## Abstract

The 1616 Program is a newly developed and evidence-informed story-based positive youth development (PYD) program for young ice hockey players (10–12 years of age) in North America. The program uses elite ice hockey players as role models—through story-telling—to serve as inspirational figures to engage youth athletes and important social agents (i.e., parents, coaches) with evidence-informed PYD concepts. The objective of this study was to use a Proof-of-Concept evaluation to assess whether the 1616 Program ‘worked’ in enhancing PYD outcomes and to determine if the concepts were engaging and enjoyable for youth, their parents, and coaches. The 5 week Proof-of-Concept evaluation was conducted with 11 ice hockey teams (*n* = 160 youths, 93 parents, and 11 coaches), encompassing both qualitative (e.g., focus groups) and quantitative (e.g., retrospective pretest-posttest questionnaires) processes and outcome assessments. Results showed that the program was well received by participants and positively impacted the intended outcomes. Overall, the data presented in this Proof-of-Concept evaluation was deemed to support the development and implementation of the full-scale 1616 Program for a more comprehensive evaluation.

## 1. Introduction

Extensive evidence supports the positive outcomes of sport participation for youth [1]. Unfortunately, despite the potential for physical, social, and emotional development, researchers caution that simply participating in sport is not enough [2]. Indeed, sport should not be assumed to automatically contribute to healthy developmental opportunities, especially with the growing emphasis on performance and achievement at younger and younger ages [3,4,5]. As such, ensuring that sport programs are age appropriate, include challenging and enjoyable activities, and involve quality social relations is paramount for promoting sport as a vehicle for positive youth development (PYD) [6]. Given calls to develop evidence-informed and practically relevant PYD programs that maximize the benefits associated with sport participation [7,8], within the following paper we describe the initial evaluation process of a novel PYD program meant to intentionally improve the sport experiences for youth, their parents/caregivers, and their coaches.

Given that youth sport programs serve as a viable avenue to promote PYD, sporting organizations have sought to develop programming that can assist in ensuring that involvement-related benefits are realized [9]. The Ladd Foundation is one such charitable organization that is set on promoting developmentally rich sport opportunities for youth involved in ice hockey across North America. Brandy and Andrew Ladd, the founders of the Ladd Foundation, assembled a team of experts to develop the 1616 Program. The program derives its name from Andrew’s National Hockey League (NHL) playing number, 16, the duration of the program, 16 weeks, and the year 1616, when the term “buffalo” was first used to describe the American Bison. The program’s main message centers around developing a ‘Buffalo Mindset,’ which focuses on banding together and moving through a challenge with the support of a herd, akin to a buffalo’s behavior.

A team of researchers was asked to assist with developing the program to ensure that the program content was evidence-informed and aligned with the sporting context. Since the primary delivery mechanism was storytelling, various other organizations specializing in educational material and digital and video content creation were engaged in the partnership (e.g., Impact Society, Banner TV, The Post Game, and Anthem). For a more detailed description of partnership development, please see Martin et al. [10]. The content of the 1616 Program provides a PYD-facilitative experience to young athletes, integrated as scaffolding for their typical ice hockey participation. It also provides parents and coaches with actionable knowledge and strategies closely related to and supportive of the activities that the athletes are engaging in. As numerous invested partners were involved throughout development, the intervention was designed following the guidelines of an integrated Knowledge Translation approach (iKT) [10,11]. A major strength of iKT projects is that they leverage an organization’s resources and unique experiences in a particular context, combined with specific topic expert knowledge, to result in more relevant and impactful policy and practice [11].

Through the 1616 Program, the Ladd Foundation sought to promote healthy sport experiences by capitalizing on inspirational stories from elite athletes without over-structuring or detracting from physical practice and playing time for youth. Their initial objective was to ensure that an intervention with youth ice hockey players did not impose additional requirements (e.g., training workshops) on already busy parents and coaches and, perhaps most importantly, was free for any interested teams/organizations. They grounded the 1616 Program in PYD and coaching literature, specifically the sport-based Personal Assets Framework (PAF) [6]. This body of literature suggests that youth thrive as a result of the dynamic interactions of developmental systems and the adaptive regulations of person-context relations [12,13]. Specifically, the PAF suggests that through the interactions of three dynamic elements (i.e., appropriate settings, quality social dynamics, and personal engagement in activities), participation can foster both short-term (i.e., across a sport season; 4Cs: competence, confidence, connection, and character) and long-term (i.e., the cumulation of multiple seasons; 3 Ps: participation, performance, and personal development) outcomes for young athletes. The main messaging throughout the program emphasizes the 4Cs, which our partnership used to represent PYD generally.

It is important to recognize that developing and evaluating novel sport interventions should not be seen as separate processes. For instance, recent systematic reviews and meta-analyses emphasize that, although there is some evidence for the effectiveness of sport-based PYD interventions, the quality of evaluations and transparency of the research process are lacking [14,15]. In recent years, researchers have seen an increase in the use of both outcome and process evaluations in youth sport programming, with evidence linking program features (i.e., delivery and functionality) as central to achieving expected outcomes [16,17]. Researchers have noted the need to avoid viewing outcomes as either attained or not attained [18,19]. Instead, they suggested focusing on substantial individual variation in conditions, capabilities, and resources linked to program settings and implementation. Thus, an essential aspect of determining the quality of sport-based PYD programs and how they can be improved and sustained over time is to understand what components make them effective, how they affect the participants, and under what circumstances [14,16].

Given the extensive personnel, time, and financial resources dedicated to the development of the 1616 Program [10], it was necessary first to evaluate how the PYD concepts were experienced and perceived by youth, parents, and coaches before continuing to conduct full-scale implementation and evaluation. Therefore, before making a final decision on the content and processes of finalizing a full 16 week 1616 program, we conducted a Proof-of-Concept (PoC) evaluation to assess the impact and effectiveness of the program on a shorter time scale and with an accessible sample [20]. Importantly, PoC trials are a resource-effective way to demonstrate (a) preliminary efficacy on target behavioral mechanisms (i.e., outcome evaluation) and (b) implementation effectiveness (i.e., process evaluation) to inform the development of more comprehensive iterations and rigorous evaluations of a program [21,22]. The purpose of our PoC, then, was to collect feedback to improve program content and delivery as well as gain certainty that the concepts and story-based intervention approach were valuable to the participants. As such, the PoC involved a 5 week evaluation of a sample of the weekly story-based content and activity structures.

## 2. PoC Purpose

Two primary research objectives were determined through several PoC collaborative planning meetings with the Ladd Foundation, Impact Society, and the research group. Objective 1 (OB1) was to evaluate whether the 1616 Program ‘worked’ in enhancing PYD-related outcomes in youth. Objective 2 (OB2) was to determine if the concepts were engaging and enjoyable for youth, their parents, and coaches. More specifically, we were interested in knowing if representative weeks from the proposed program could positively impact youth competence, confidence, connection, and character (i.e., outcome evaluation; OB1) and whether the program’s PYD concepts and story-based structure were feasible and acceptable—that is, appropriate, delivered effectively and engagingly, and whether participants were satisfied with the quality (i.e., process evaluation; OB2). Ultimately, this study’s objective was to gather information from the participants that would help inform further decisions about full-scale program development and implementation.

## 3. Methods

### 3.1. PoC Program Overview

For each week during the 5 week PoC evaluation, new material was introduced to youth, parents, and coaches on a selected topic from existing sport and PYD literature (i.e., spanning confidence, connection, or character). The primary (and unique) content delivery mechanism involved a ~5 min edited video of a professional (e.g., NHL, Professional Women’s Hockey Players’ Association) or international (e.g., Olympic) ice hockey player telling their personal story concerning a particular PYD concept. As an overview of the PoC content, week 1 involved a program introduction focused on the concept of commitment/engagement. Weeks 2–4 involved the concepts of morality/integrity, psychological safety, and self-efficacy, followed by a consolidation and program conclusion in week 5. Throughout the program, participants were given access to online videos/stories in addition to reflection and action-based activities (e.g., ‘Live It Outs’) each week. The videos/stories, and additional activities were narrated and introduced by an animated buffalo named ‘Buffalou’. The program’s content was tailored for relevance to young ice hockey players aged 10–12 years.

In addition to the content provided to youth, parents were sent resources each week to support their child through the sport experience (e.g., ‘the Car Ride Home’, ‘Conversation Starters’). Importantly, these resources were informed by current youth sports literature about parent education, communication, and reflective practice [23,24,25]. Similarly, coaches were sent coaching-specific tips and ice hockey-specific drills to include within their practices—all aimed at improving their professional, interpersonal, and intrapersonal knowledge and behaviors [26,27].

### 3.2. Participants

The PoC was conducted with a convenience sample comparable to previously completed mixed-methods PoC studies exploring novel behavioral interventions [28]. Samples for PoC testing can be small and accessible rather than large and representative, given the purpose of determining whether continued development and more rigorous testing are merited [21]. Participants were recruited through word-of-mouth and snowball techniques led by the partner organization. In total, 11 youth ice hockey teams from U9 to U14 age groups, including their parents and coaches, from across North America were involved. Although the program was designed for the 10–12 year-old range, the age range was expanded to explore receptiveness with younger (9 year-olds) and older (14 year-olds) populations. Five teams were from Canadian provinces (British Columbia, Alberta, Manitoba, and Ontario), and the remaining six were located across the United States (Illinois, Maryland, and North Carolina). Whereas the overall participant pool consisted of 11 teams with parents and coaches, not all invested partners from each team provided data (e.g., questionnaire/interview responses) for the PoC. Below, we provide specific sample demographics pertaining to responses for the different evaluation components.

### 3.3. Evaluation Design and Analysis

Given the aim of quality assurance and program improvement from the PoC for the Ladd Foundation, this project received approval from the first author’s institutional research ethics board. Specific quantitative and qualitative methods were selected to balance the need to satisfy our research objectives while remaining practical and user-friendly for participants—a priority outlined by the Ladd Foundation and the iKT approach in general. First, the research team used questionnaires (e.g., retrospective pretest-posttest [RPP] and one-group pretest-posttest questionnaires) to assess changes in youth, parents, and coaches’ desired outcomes (OB1). It is important to highlight that, unlike traditional pretest-posttest questionnaires, RPP has respondents, at *one* posttest time point, rate items based on their recall of two specific instances: ‘now’ and ‘before’ [29]. This decision was made in consultation with the partners, who emphasized the need to limit the time demands imposed on participants. In partnership research, it is critical to consider the requests of all partners when attempting to achieve common objectives [30]. It is also worth noting that an RPP approach allows participants to more accurately gauge the degree of change between time points through greater self-awareness [29]. All quantitative measures were collected remotely via SurveyMonkey. Second, focus group interviews were conducted at the end of the 5 week program to explore engagement and enjoyment for youth, their parents, and coaches (OB2). The interviews were conducted via Zoom, recorded, and transcribed with participant permission, then thematically analyzed [31]. Recurring themes, suggestions, and feedback were summarized concerning each invested partner group.

The following sections outline the methods used to assess each objective, separated for each partner. As such, the first section introduces and describes the methods used to evaluate selected outcomes (OB1) for youth, parents, and coaches. The subsequent section addresses methods used to assess process-related factors (OB2) for youth, parents, and coaches.

### 3.4. Outcome Evaluations (OB1)

#### 3.4.1. Youth

**Retrospective Pretest-Posttest (RPP) Weekly Questionnaires.** In RPP format, youth from the 11 teams provided weekly information on learning and the impact of the weekly program content. The weekly questionnaires targeted perceptions of the story topic that week and were completed based on a Likert scale from 1 (*not at all*) to 5 (*very much*) for two time points: ‘now’ and ‘before this week’. Week 1 included items from the commitment (4 items, e.g., ‘How dedicated are you to playing hockey on this team?’) and enjoyment (4 items, e.g., ‘Do you enjoy playing hockey this season?’) subscales from the sport commitment model [32]. Week 2 included items from the Youth Sport Values Questionnaire (YSVQ-2) for the moral (3 items, e.g., ‘I try to be fair’), competence (3 items, e.g., ‘I set my own targets’), and status (3 items, e.g., ‘I am a leader in the group’) subscales [33]. Week 3 included items meant to assess perceptions of a psychologically safe climate pertaining to coach support (4 items, e.g., ‘My coach is flexible about how I play my position’), role clarity (3 items; ‘The amount expected of me is clearly defined’), and self-expression (3 items; ‘It is okay to express my true feelings’) previously used in sport [34,35]. Week 4 had five items assessing sport self-confidence (e.g., ‘I am confident about performing well in hockey’) from the Competitive State Anxiety Inventory-2 [36]. Finally, week 5 included four items, representing a composite measure of the Cs (e.g., connection: ‘I feel connected to teammates’). From the 11 hockey teams, an average of 63.40 (*SD* = 13.8) youth (*M_age_* = 10.90 years, *SD* = 0.19) completed the questionnaire every week.

**Pretest-Posttest Questionnaires.** A pretest-posttest questionnaire was created and administered before week 1 and at the end of week 5. Importantly, this decision represented the first stage of an iterative process whereby the results were deemed secondary. We recognized that not all topics would be covered in five weeks; however, given that the eventual complete program would include a pre-post assessment, we were interested in exploring the practicality, feasibility, and receptiveness of questionnaire completion. In collaboration with the partners, item decisions were made to provide a comprehensive evaluation within a manageable time for completion (~20–30 min). Following guidelines advanced by Vierimaa et al. [37] for creating a PYD Cs ‘Tool Kit,’ 70 items expected to represent connection [32,38,39], confidence [36], and character [40,41] were selected, and responses were provided on a Likert scale from 1 (*not at all/none/nothing*) to 5 (*very much/a lot*). Across the 11 hockey teams, 160 youth (44% female; *M_age_* = 11.17 years; *SD* = 4.30 years) responded to the pretest questionnaire, and 83 (*M_age_* = 10.8 years; *SD* = 1.20 years) responded to the posttest questionnaire. Across the sample, 82% identified as white, 9% as Eastern Asian, 6% preferred not to answer, 1% as black or African American, 1% as Middle Eastern, and 1% as race/ethnicity not listed.

**Follow-up Focus Group Interviews.** Using interviews alongside small sample pre-experimental studies can assist in determining whether to move forward with program development and rigorous testing [21]. Following the 5 week PoC, focus groups (*n* = 4) were conducted with 14 youth aged 9 years (*SD* = 1.25 years) to explore engagement and learning through the 1616 Program (e.g., ‘What did you learn from the story that the player told?’).

#### 3.4.2. Parents

**RPP Questionnaires.** Parents (*n =* 93; 46% female) participated in a post-program questionnaire presented in RPP format meant to compare perceptions about (a) their child, (b) themselves, and (c) the sport environment before and after the program. Four items about their children were specific to each ‘C’ (e.g., connection: ‘I believe that my child feels connected to their teammates’). Twenty items about themselves were based on the COM-B behavior change theory [42,43] to explore their perceived *Capability* (e.g., ‘I have the necessary knowledge and skills to support my child’s confidence’), *Opportunity* (e.g., ‘I am aware of opportunities where I can support my child’s confidence’), and *Motivation* (e.g., ‘I am motivated to support my child’s confidence’) to engage in *Behaviors* that supported their child and about their sport parent self-efficacy (e.g., ‘How confident are you in your ability to promote good sportspersonship’) on a Likert scale ranging from 1 (*not at all*) to 5 (*very much*) [44]. Finally, we were also interested in knowing whether parents felt that the program improved the quality of the sport environment by adapting 41 of the 51 items from the Program Quality Assessment in Youth Sport (PQAYS) to a self-report format for psychological safety, appropriate structure, supportive relationships, opportunities to belong, positive social norms, support for efficacy and mattering, opportunities for skill-building (sport/physical and life skills), and integration of family [45]. On average, parents were 42.5 years old (*SD* = 6.67), and 81% identified as white, 13% as Eastern Asian, 3% preferred not to answer, 2% had no race/ethnicity listed, and 1% were black or African American.

**Follow-up Focus Group Interviews.** At the end of the 5 weeks, a sample of willing parents (*n* = 13; 23% female) took part in remote (i.e., Zoom) focus groups (*n =* 5) about their experiences, perceptions, and engagement with the program. To understand parent-related outcomes, parents were asked questions such as, ‘Did you learn anything from the program?’ and ‘What aspects of the program had the greatest impact on you?’.

#### 3.4.3. Coaches

**RPP Questionnaires.** One coach (head or assistant) from each of the 11 teams completed an RPP post-program questionnaire meant to explore changes in perceptions before and after the program about (a) their athletes and (b) themselves. Similar to parents, coaches rated their athletes’ improvement for the Cs based on an item for each (e.g., connection: ‘I believe that my athletes feel connected to their teammates’). They also completed items about their perceived *Capability*, *Opportunity*, and *Motivation* to engage in *Behaviors* that supported their athletes [42,43] and about their coach-specific efficacy [44] with items adapted to coaching roles (e.g., ‘How confident are you in your ability to build team cohesion’) on a Likert scale ranging from 1 (*not at all*) to 5 (*very much*). All coaches (*n =* 11) were identified as male and were, on average, 44.82 years old (*SD* = 7.39) and predominantly white (9% race not identified).

**Follow-up Focus Group Interviews.** A sample of coaches (*n* = 8) took part in follow-up interviews to explore their experiences, perceptions of, and engagement with the program. To explore coach-related outcomes, coaches were asked questions such as, ‘What plans do you have to change anything or do anything differently as a result of this program?’ and ‘What aspects of the program had the greatest impact on you?’.

### 3.5. Process Evaluations (OB2)

#### 3.5.1. Youth

**Weekly Process Questionnaires.** At the end of each week, youths were asked ten process-related questions that were responded to with ‘yes’ or ‘no’ or on a Likert scale from 1 (*not at all*) to 5 (*very much*). Four items were experiential in nature (e.g., ‘Did you have fun this week?’) and six were engagement-related (e.g., ‘Did you watch the video this week?’). Overall, these items were meant to determine how enjoyable the weekly video was, the degree to which the reflection items helped them understand the topic, and if the ‘Live It Out’ action item was useful.

**Posttest Process Questionnaires.** At the end of the program, the youth responded to seven items addressing program-specific feedback about the quality and amount of content shared (e.g., ‘Did Buffalou add value to the videos?’; ‘What did you think of the amount of stuff we shared?’; ‘How was the length of the weekly player videos?’). These were responded to by selecting a provided option or were rated on a scale from 1 (*not at all*) to 5 (*very much*).

**Follow-up Focus Group Interviews.** During the focus groups discussed previously for OB1, youth were also asked more process-related questions. These were meant to target overall program implementation and experience (e.g., ‘Did you like receiving the different stories online through videos?’; ‘Which video did you like/dislike the most and why?’).

#### 3.5.2. Parents

**Post-Program Process Questionnaires.** Fifteen items were provided to parents addressing the quality of the content and delivery and descriptions of intentions for future use (e.g., ‘The 1616 Program was well structured and organized’; ‘I would recommend the 1616 Program to others’) that were answered on a 7-point Likert scale ranging from 1 (*strongly disagree*) to 7 (*strongly agree*).

**Follow-up Focus Group Interviews.** During the OB1 interviews discussed previously, parents were also asked about their perceptions of how the program was delivered, what they thought about its quality, and if they found it enjoyable (e.g., ‘What did you think about the way the content was delivered?’).

#### 3.5.3. Coaches

**Post-Program Process Questionnaires.** Fifteen items were given to coaches that sought feedback about the quality of delivery and intentions for future use (e.g., ‘The 1616 Program was well structured and organized’; ‘I would recommend the 1616 Program to others’) on a 7-point Likert scale from 1 (*strongly disagree*) to 7 (*strongly agree*).

**Follow-up Focus Group Interviews.** In addition to the questions discussed previously for OB1, more in-depth feedback about the delivery, quality, and enjoyment (e.g., ‘What did you think about the way the content was delivered?’) was obtained in the post-program follow-up interviews with coaches.

## 4. Results

### 4.1. Outcome Evaluations (OB1)

#### 4.1.1. Youth

**RPP Weekly Questionnaires.** Table 1 provides the means (*SD*), mean difference, *t*-statistic, *p*-value, and confidence interval (CI) for the weekly topics assessed. Athletes reported a significant difference in perceptions for both commitment (M*diff*: 0.12, *p* < 0.001, CI: [0.05, 0.18]) and enjoyment (M*diff*: 0.06, *p* < 0.001, CI: [0.02, 0.11]) during week 1. Moral values (M*diff*: 0.17, *p* < 0.001, CI: [0.09, 0.24]) and competence values (M*diff*: 0.21, *p* < 0.001, CI: [0.11, 0.30]) demonstrated a significant difference for week 2, while role clarity (M*diff*: 0.09, *p* = 0.002, CI: [0.04, 0.15]) demonstrated a significant difference for week 3. Perceptions of confidence (M*diff*: 0.29, *p* < 0.001, CI: [0.18, 0.41]) increased significantly during week 4, as did the PYD Cs (M*diff*: 0.33, *p* < 0.001, CI: [0.22, 0.45]) during week 5.

**Pretest-Posttest Questionnaires.** Table 2 provides the means (*SD*), mean difference, *t*-statistic, *p*-value, and CI for the pre- and post-topics assessed. Overall, measures remained unchanged or improved marginally. Increases in youth cognitive centrality (M*diff*: 0.20, *p* = 0.059, CI: [−0.01, 0.40]) and confidence (M*diff*: 0.12, *p* = 0.061, CI: [−0.01, 0.25]) were observed. Only mental toughness (M*diff*: 0.21, *p* = 0.002, CI: [0.08, 0.33]) demonstrated a significant difference.

**Follow-up Focus Group Interviews.** Youth engagement and learning were evident from follow-up focus groups. As an example, athletes described learning to take responsibility for their development in hockey and showed an understanding of how their work ethic would benefit their competence and confidence: ‘I learned to try my hardest, even when the coaches or your parents aren’t looking and, to be honest, always’ (Player 12, U9). There was also some indication that the messages learned could translate beyond the hockey setting: ‘It was fun, I actually learned about how to be more confident in myself” (Player 13, U9), and ‘Being confident in yourself in school like if you have a test, and at hockey’ (Player 5, U9).

#### 4.1.2. Parents

**RPP Questionnaires.** Table 3 provides the means (*SD*), mean difference, *t*-statistic, *p*-value, and CI for the RPP responses. A significant difference was observed in parents’ perceptions of improvement in their children’s 4Cs (*Mdiff:* 0.54, *p* < 0.001, CI: [0.43, 0.65]). There were also significant differences in perceptions of their own capability (M*diff*: 0.57, *p* < 0.001, CI: [0.42, 0.71]), opportunity (M*diff*: 0.61, *p* < 0.001, CI: [0.47, 0.75]), and motivation (M*diff*: 0.42, *p* < 0.001, CI: [0.29, 0.56]) to support their child. Parents also demonstrated significant differences in their perceptions of their efficacy toward motivating their children (M*diff*: 0.55, *p* < 0.001, CI: [0.42, 0.68]) and character building with their children (M*diff*: 0.29, *p* < 0.001, CI: [0.19, 0.40]). Importantly, significant differences for all the features of a quality sport environment were observed (see Table 3).

**Follow-up Focus Group Interviews.** These results were further supported and contextualized by interview responses whereby parents discussed how the program influenced their general ability to support their children. For instance, one participant stated in relation to their capability and opportunity: ‘We gravitated towards the conversation starters…we’re always trying to get something started with our son because he’s all over the place…it was nice to get him started on a path to have a discussion’ (Parent 1, U10). Parents also commented on how the program shaped their team’s sporting environment:

‘Our team hasn’t won a game yet, so there’s been lots of things that haven’t gone well…this has been a very good program for our team. We’ve all faced adversity, so we talk about going into the storm and about being a herd. It’s been very timely for our group. I think it’s helped, pardon the pun, ‘weather the storm’’. (Parent 4, U13).

#### 4.1.3. Coaches

**RPP Questionnaires.** Significant improvements were observed in all but one of the topics assessed with coaches (see Table 4). They felt that their athletes improved for the 4Cs (M*diff*: 0.91, *p* < 0.001, CI: [0.57, 1.25]) and they felt they themselves improved in their capability (M*diff*: 0.59, *p* = 0.003, CI: [0.24, 0.94]), awareness of opportunities (M*diff*: 0.68, *p* = 0.006, CI: [0.24, 1.12]), and motivation (M*diff*: 0.86, *p* < 0.001, CI: [0.48, 1.25]) to support their athletes. They also improved in their beliefs for character building (M*diff*: 0.39, *p* = 0.002, CI: [0.17, 0.62]), but not for their ability to improve hockey technique.

**Follow-up Focus Group Interviews.** Coach responses supported the previous results during interviews. For example, a coach discussed how the program allowed them to support athletes’ development of the Cs: ‘It gave us coaches an idea every week of different things to try with your group; whether it’s character building or on ice—actual skill-specific stuff. It’s an [in-depth] resource…overall just such a positive experience for everyone’ (Coach 4, U10). Coaches also commented on how the program enhanced their motivation to support their athletes: ‘The videos really spoke a lot to me as a coach. I want to make that difference now. I don’t want a player to have success in spite of me’ (Coach 2, U14).

### 4.2. Process Evaluations (OB2)

#### 4.2.1. Youth

**Weekly Youth Process Questionnaires.** As shown in Table 5, process features about perceptions of enjoyment, learning, improvement, and excitement were highly rated. From a fidelity perspective, it is also clear that almost all participants reported watching the videos, completing the reflection questions, and engaging with the ‘Live it Out’ activities. Interestingly, differences in engagement could be seen when broken down based on age group (i.e., U10, U12, and U14). Specifically, 89% of U10 youth and 86% of U12 youth rated their level of fun engaging in the program as a four or higher. However, as youth age increased, the description of fun decreased. For instance, only 64% of U14 youth rated their level of fun as a four or higher. A similar finding was observed for the level of excitement to engage with the full 1616 Program. Whereas 86% of U10 youth rated excitement as a four or higher, 73% of U14 youth rated their level as a four or higher. In relation to the quality of the weekly video, 95% of U10 and U12 youth selected four or above, compared to 60% of U14 youth. Comparable ratings were observed when youths were asked to rate their engagement with the reflections and LIOs.

**Post-Program Process Questionnaires.** Table 6 demonstrates that generally, the youth felt strongly about the quality and content of the resources and identified their favorite videos from the program.

**Follow-up Focus Group Interviews.** Overall, there was a sense of engagement with the stories being told, and athletes could relate to the messages delivered by the players. For instance, they described their excitement with the way the reports were delivered: ‘It was fun to see different players and the stories that they’ve been through’ (Player 1, U9) and ‘…it felt like you actually knew them personally’ (Player 5, U9). However, focus groups also highlighted the need for content to be age appropriate. Similar to the results presented above, older athletes were not as engaged with certain aspects, such as the buffalo mascot for the program, ‘I didn’t like the buffalo in the first two videos. He was a little much’ (Player 14, U14). Conversely, younger athletes found the pace and some of the content difficult to follow. For instance, although athletes described enjoying the stories and self-reflection questions, one athlete discussed needing more clarity on the weekly messages, ‘Just the way it’s said could be a little better… I guess [the message] just has to be a little clearer’ (Player 1, U9). Additionally, another player discussed the fast pace of the weekly content, ‘Maybe give us more time to talk about [the content of] each week’ (Player 6, U9).

#### 4.2.2. Parents

**Post-Program Process Questionnaires.** Table 7 shows that on a Likert scale from 1 (*strongly disagree*) to 7 (*strongly agree*), 100% of parents selected a response of five or higher when describing the program as aligning with their beliefs and values, as useful, credible, and a positive experience, as well-structured and organized, and as effective. Areas of improvement marked by parents for the Ladd Foundation to consider were observed in relation to effectiveness over other resources (82%), the available evidence of the impact of the program (91%), and time requirements (91%).

**Follow-up Focus Group Interviews.** During interviews, parents expanded their appreciation for the resources shared through the program. One parent elaborated: ‘The ‘Chats with Brandy’ videos where you can see them talking to their kids and how they’re relaying the same weekly messages to them in their own day-to-day life…that gave us more to talk about than just the normal ‘How was your day?’ The way that it tied a lot of things together was really beneficial.’ (U10 Parent).

Parents also highlighted several challenges/barriers they experienced with the program, including the frequency of messaging they were receiving and the speed at which new content was being presented: ‘I feel like it was a lot of texts and emails. I would say ask people what they prefer and then just get that one [form of communication] rather than both’ (U10 Parent). Another parent recommended spacing out the delivery of the content to allow for more time to digest the messaging: ‘We were talking about spreading the [content] out… then they really get to dive a little bit deeper into each one of them’ (U10 Parent).

#### 4.2.3. Coaches

**Post-Program Process Questionnaires.** Table 8 shows that on a Likert scale from 1 (*strongly disagree*) to 7 (*strongly agree*), 100% of coaches selected a response of five or higher to describe the program as aligning with their beliefs and values, as useful, credible, and a positive experience, as well-structured and organized, and as effective. Whereas all items were rated highly, areas to consider for improvement could be demonstrating effectiveness over other resources (82% five or higher), available evidence of the impact of the program (91% five or higher), and time requirements (91% five or higher).

**Follow-up Focus Group Interviews.** Coaches provided contextualizing feedback during the follow-up interviews: ‘The 4 Cs…that’s the stuff that carries on and off the ice, and if hockey becomes a teaching tool for that, great. That’s where the strength of the program is, the culture. The 4 Cs did resonate with the [athletes]’ (Coach 3, U12). Coaches also discussed witnessing the program’s positive impact on their athletes: ‘It’s a positive experience for the kids. They really start to learn about overcoming adversity, struggling, and character building, and it’s presented in a great way’ (Coach 4, U10).

## 5. Discussion

The purpose of this article was to describe the evaluation of a PoC trial involving the 1616 Program. The two overarching objectives for the 5 week PoC were to evaluate whether the 1616 Program ‘worked’ in enhancing PYD-related outcomes, validate the story-based intervention approach (i.e., OB1), and explore participant process-related experiences (i.e., OB2). In doing so, we would determine whether full-scale development was warranted and what changes needed to be made before rolling out the complete program. Within the following sections, we summarize the findings, discuss implications as they pertain to the literature, and conclude by describing the next steps for continued development and refinement of the program.

### 5.1. Outcome Evaluation (OB1)

There are relevant outcome-related findings for youth, parents, and coaches. Concerning youth, the RPP evaluations demonstrated that every week, the youth felt that they improved in relation to the topic introduced. Given that the eventual 1616 Program will aim to include a weekly message over 16 weeks, this is a promising finding. In other words, an introductory video/story, accompanied by a reflection question and ‘Live It Out’ activity, represents a format whereby youth feel like they learn and improve with a particular PYD topic. Although the theoretical and conceptual foundation and rationale for the program structure are described elsewhere [10], we expected that introducing significant role models and reflection questions could leverage social learning and behavior modeling theories for youth [46]. Similarly, we also considered goal setting and action planning research [47,48] in relation to how the ‘Live It Out’ activities would enable youth to engage in relevant behaviors that aligned with the topic and were meaningful to them. From a measurement perspective, it is also worth noting the benefits of using RPP formats for variables that traditionally have high baseline measures [49], which was found with the current sample.

Interestingly, the only significant finding concerning the pretest-posttest questionnaires involved mental toughness [40]. Although a representative dimension of social identity and confidence was nearing significance, this finding warrants discussion. While many reasons could be advanced, two in particular should be noted. First, as seen in Table 2, it is possible that a ceiling effect was present, with participants leaving little room for improvement at Time 2. This finding supports our decision to include pretest-posttest and RPP evaluations [50]. Second, the 5 week PoC structure did not represent the theoretically informed planning for the full 16 week program [10]. For instance, concepts for connection were not directly included due to timing and athlete availability decisions, so it is perhaps not surprising that those specific variables did not improve. As the eventual program is finalized, it will be imperative to ensure that the variables and measurements selected align with our conceptualization of the topics covered and that these are clearly articulated [14]. Similarly, whereas the eventual program will involve a 1 week introductory/engagement phase, a 13 week ‘Cs’ promotion phase, and a 2 week consolidation/maintenance phase grounded in team building intervention literature [51], this was not possible in the PoC, and so the weekly content represented more of a silo structure. Therefore, despite the lack of significance from the pretest-posttest evaluations, we believe that for the reasons stated here, in conjunction with the findings from the RPP and the follow-up interviews, youth generally benefited from the program.

Regarding parents, results suggested favorable outcomes. For instance, in comparison to before the program, they believed their children improved across the 4Cs. Further, the quality of the sport environment is critical for healthy development [6], and parents reported significant increases across all dimensions proposed by Bean et al. [45]. They also reported improvements in beliefs about their capability, awareness of opportunities, and motivation to support their children, which are important indicators for eventual engagement in behaviors [43]. Finally, the importance of self-efficacy is widely accepted [46], so it is noteworthy that the content provided to parents improved their confidence in their own parenting/ability to support their children. Through the interviews, parents noted how the resources helped them engage in conversations with their children that they might not have otherwise had. They also discussed how their conversations extended beyond ice hockey, whereby they implemented the tips with their child(ren) in various contexts, such as school.

Our results suggested that the program was also effective from the coach’s perspective. For instance, they felt that their athletes improved for the 4Cs. Similarly, they felt that their capability and motivation to support their athletes improved, in addition to their awareness of opportunities to do so [43]. It is worth noting that although they also had stronger beliefs in their ability to promote character building amongst their athletes, the program was not felt to have improved their ability to teach hockey techniques. Although the main aim of the program is to emphasize PYD generally, one of the Cs is competence, so helping coaches improve hockey skills specifically is important. As we continue with program development and plan a more comprehensive evaluation of the full program, we will need to explore how the skill videos provided to coaches can more effectively influence their efficacy beliefs. Finally, in addition to these findings, coaches discussed how the resources helped them avoid ‘tunnel vision’ towards performance-related outcomes and instead promoted intentional behaviors and helped them understand the impact of their actions on their athletes.

### 5.2. Process Evaluation (OB2)

The second aim of the PoC evaluation was to explore participant experiences with the program. Our findings suggest that the overarching idea and messaging of the 1616 Program were well received by all participants. For example, focus group discussions with each of the stakeholders revealed that the ‘Buffalo Mindset’ and 4Cs resonated and aligned with participants’ sport values. For the youth specifically, almost all reported engaging with the content, and the mean responses were all above four on a 5-point Likert scale when asked if they were enjoyable, if they improved understanding, and if they were worthwhile (see Table 5). Similarly, and based on the planning process for the PoC, the feedback was requested by and noted as particularly impactful for the creative committee from the iKT partnership [10] in relation to the amount of content provided and the length of the videos, as two examples.

The coaches and parents appreciated the common language that the program provided and how it was applicable to other contexts, providing them with a language/vocabulary to communicate with their athletes and children. Suggestions for improvement from these participants mainly focused on program delivery rather than the content itself. For instance, they indicated that the content and messaging were being sent too frequently. As such, in subsequent conversations with the program partners, the quality and intentionality of content delivery will be prioritized for the proposed 16 week program. This will allow time for content to be absorbed by participants and provide opportunities for greater engagement in activities. Notably, the lowest responses from coach and parent feedback were their perceptions that (a) the 1616 Program was more effective than other programs, (b) evidence of its potential impact was available, and (c) time requirements were appropriate (see Table 7 and Table 8). This finding aligns with research on parent education programs in sport, where parents noted that “knowledge is power…” [52] (p. 441). In this regard, the onus will be on future iterations of the program to ensure that parent resources are informative and practical and that the evidence from which the suggestions are taken is readily available to parents. Similarly, the time requirements will be a critical consideration, as time is a consistent barrier to coach and parent programs [49]. This has been an a priori objective for the Ladd Foundation since program development’s inception.

## 6. Conclusions

This article described the initial evaluation of a 5 week PoC test of the 1616 Program. This was an important first step in program development and represented a novel attempt to evaluate an initial program originating from an iKT approach in sport. Indeed, recent reviews highlight the limitations of PYD interventions in sports [14]. Thus, the descriptive account of our evaluation highlights the benefits of engaging in PoC testing of both outcome and process evaluations [16] and could serve as a useful template for others interested in such undertakings. Notably, although the findings from both the process and outcome evaluations justify continued program development and implementation of the full 16 week program, this PoC evaluation highlights important considerations and justifies changes to ensure that the program is most impactful for youth.

## Figures and Tables

**Table 1 children-10-00799-t001:** Youth means, standard deviations, mean difference, *t*-statistic, *p*-value, and confidence interval for RPP responses.

Variable Name	Mean T1 (*SD*)	Mean T2 (*SD*)	Mean Difference	*t*-Statistic	*p*-Value	95% CI
Commitment	4.71 (0.24)	4.83 (0.31)	0.12	3.70	<0.001	[0.05, 0.18]
Enjoyment	4.79 (0.36)	4.85 (0.32)	0.06	2.64	<0.001	[0.02, 0.11]
Confidence	4.04 (0.68)	4.33 (0.62)	0.29	5.20	<0.001	[0.18, 0.41]
Moral values	4.69 (0.38)	4.86 (0.22)	0.17	4.47	<0.001	[0.09, 0.24]
Competence values	4.6 (0.52)	4.81 (0.35)	0.21	4.40	<0.001	[0.11, 0.30]
Status values	3.57 (0.88)	3.65 (0.90)	0.07	1.81	0.075	[−0.01, 0.15]
Psych safety—Coach support	4.06 (0.70)	4.1 (0.68)	0.04	1.66	0.101	[−0.01, 0.09]
Psych safety—Role clarity	4.36 (0.59)	4.45 (0.54)	0.09	3.28	0.002	[0.04, 0.15]
Psych safety—Self expression	4.35 (0.66)	4.41 (0.65)	0.05	1.75	0.085	[−0.01, 0.11]
PYD Cs	4.17 (0.61)	4.5 (0.48)	0.33	5.95	<0.001	[0.22, 0.45]

**Table 2 children-10-00799-t002:** Youth means, standard deviations, mean difference, *t*-statistic, *p*-value, and confidence interval for pre- and post-responses.

Variable Name	Mean T1 (*SD*)	Mean T2 (*SD*)	Mean Difference	*t*-Statistic	*p*-Value	95% CI
Commitment	4.79 (0.36)	4.82 (0.41)	0.06	1.02	0.312	[−0.05, 0.16]
Enjoyment	4.85 (0.34)	4.83 (0.41)	0.02	0.30	0.762	[−0.09, 0.11]
Confidence	4.28 (0.60)	4.9 (0.48)	0.12	1.91	0.061	[−0.01, 0.25]
Task-orientation	4.63 (0.40)	4.65 (0.39)	0.02	0.42	0.678	[−0.09, 0.14]
Ego-orientation	3.44 (1.00)	3.29 (1.15)	0.02	0.21	0.837	[−0.21, 0.26]
Ingroup ties	4.45 (0.71)	4.44 (0.79)	0.13	1.57	0.122	[−0.03, 0.28]
Cognitive centrality	4.07 (0.84)	4.12 (0.98)	0.20	1.93	0.059	[−0.01, 0.40]
Ingroup affect	4.80 (0.47)	4.68 (0.58)	−0.08	−1.65	0.104	[−0.17, 0.02]
Coach closeness	4.81 (0.37)	4.80 (0.45)	0.00	0.00	1.00	[−0.10, 0.10]
Positive parental involvement	4.64 (0.46)	4.63 (0.57)	0.04	0.46	0.650	[−0.13, 0.21]
Mental toughness	4.14 (0.50)	4.29 (0.51)	0.21	3.32	0.002	[0.08, 0.33]

**Table 3 children-10-00799-t003:** Parent means, standard deviations, mean difference, *t*-statistic, *p*-value, and confidence interval for RPP responses.

Variable Name	Mean T1	Mean T2	Mean Difference	*t*-Statistic	*p*-Value	95% CI
4Cs	3.88 (0.62)	4.42 (0.40)	0.54	9.94	<0.001	[0.43, 0.65]
Capability to support a child	3.78 (0.78)	4.35 (0.56)	0.57	7.91	<0.001	[0.42, 0.71]
Opportunities to support a child	3.66 (0.79)	4.27 (0.59)	0.61	8.54	<0.001	[0.47, 0.75]
Motivation to support a child	4.33 (0.74)	4.75 (0.41)	0.42	6.26	<0.001	[0.29, 0.56]
Sport parent efficacy—motivation	3.83 (0.76)	4.38 (0.57)	0.55	8.31	<0.001	[0.42, 0.68]
Sport parent efficacy—character building	4.50 (0.63)	4.80 (0.32)	0.29	5.49	<0.001	[0.19, 0.40]
Psychological safety created by coaches	4.23 (0.66)	4.43 (0.61)	0.20	5.44	<0.001	[0.13, 0.28]
Appropriate structure	4.59 (0.40)	4.72 (0.35)	0.13	5.08	<0.001	[0.08, 0.18]
Supportive relationships	4.31 (0.56)	4.53 (0.51)	0.22	5.76	<0.001	[0.15, 0.30]
Opportunities to belong	4.05 (0.71)	4.49 (0.62)	0.43	7.67	<0.001	[0.32, 0.55]
Positive social norms	4.59 (0.48)	4.72 (0.39)	0.13	3.75	<0.001	[0.06, 0.20]
Support for efficacy and mattering	4.08 (0.63)	4.29 (0.61)	0.21	5.70	<0.001	[0.14, 0.29]
Opportunities for skill-building: sports and physical skills	4.49 (0.48)	4.66 (0.39)	0.17	4.96	<0.001	[0.10, 0.24]
Opportunities for skill-building: life skills	3.69 (0.86)	3.93 (0.85)	0.24	5.04	<0.001	[0.14, 0.33]
Integration of families	4.59 (0.47)	4.69 (0.44)	0.10	3.09	0.002	[0.03, 0.16]

**Table 4 children-10-00799-t004:** Coach means, standard deviations, mean difference, *t*-statistic, *p*-value, and confidence interval for RPP responses.

Variable Name	Mean T1	Mean T2	Mean Difference	*t*-Statistic	*p*-Value	95% CI
4Cs	3.20 (0.60)	4.11 (0.39)	0.91	5.99	<0.001	[0.57, 1.25]
Capability to support	3.84 (0.66)	4.43 (0.39)	0.59	3.80	0.003	[0.24, 0.94]
Opportunities to support	3.75 (0.64)	4.43 (0.67)	0.68	3.46	0.006	[0.24, 1.12]
Motivation to support	3.86 (0.61)	4.73 (0.42)	0.86	5.04	<0.001	[0.48, 1.25]
Coach efficacy—motivate athletes	4.11 (0.50)	4.50 (0.48)	0.39	3.89	0.002	[0.17, 0.62]
Coach efficacy—technique development	4.62 (0.44)	4.67 (0.43)	0.05	0.54	0.602	[−0.14, 0.23]

**Table 5 children-10-00799-t005:** Question, mean (SD), and % responses above four for Weekly Youth Process Questionnaires.

Question Type	Question	Week 1 Commitment	Week 2 Morality/Integrity	Week 3 Psych Safety	Week 4 Self-Efficacy	Week 5 Cs Summary
General Experience Questions	1. Did you have fun?	4.6 (0.6)	4.6 (0.7)	4.7 (0.6)	4.5 (0.8)	4.8 (0.4)
	2. Did you learn something about yourself?	3.9 (1.2)	4.2 (0.9)	4.1 (0.8)	3.9 (1.0)	4.4 (0.8)
	3. Did you improve at hockey?	4.3 (0.8)	4.2 (0.8)	4.3 (0.9)	4.2 (0.9)	4.4 (0.8)
	4. Are you excited about next week?	4.4 (0.8)	4.3 (0.7)	4.4 (0.7)	4.3 (0.8)	-
Process/Engagement Questions	1. Did you watch the video?	99% yes	100% yes	100% yes	100% yes	98% yes
	2. How enjoyable was it?	4.1 (0.6)	4.1 (0.7)	4.2 (0.6)	4.2 (0.8)	4.4 (0.7)
	3. Did you read the reflection item?	96% yes	96% yes	97% yes	95% yes	93% yes
	4. Did it help you understand?	4.3 (0.8)	4.2 (0.9)	4.5 (0.6)	4.2 (1.0)	4.3 (1.0)
	5. Did you complete the live it out?	86% yes	91% yes	96% yes	92% yes	97% yes
	6. Was it worth it?	4.1 (0.9)	4 (1.0)	4.3 (0.7)	4.2 (0.9)	4.4 (0.7)

Note. These items were determined in consultation with the partners.

**Table 6 children-10-00799-t006:** Youth post-program process evaluation items and responses.

Feedback Item	Total Youth (*n* = 69)
1. Did you like Buffalou? (1 = Not at all/5 = Very much)	4.2 (1.1)
2. Did Buffalou add value to the videos? (1 = Not at all/5 = Very much)	4.1 (1.1)
3. Did you think the stuff with Buffalou was funny? (1 = Not at all/5 = Very much)	3.7 (1.3)
4. Did you think the stuff in the 1616 Program was cool? (1 = Not at all/5 = Very much)	4.3 (0.8)
5. What did you think about the amount of stuff we shared? (1 = Too much/2 = Not enough/3 = Just right)	7.1% (Too much) 11.4% (Not enough) 81.4% (Just right)
6. How was the length of the weekly player videos? (1 = Too much/2 = Not enough/3 = Just right)	10.1% (Too much) 7.2% (Not enough) 82.6% (Just right)
7. Which was your favorite video?	29%—Week 5 25%—Week 1 23%—Week 4 13%—Week 3 10%—Week 2

Note. These items were determined in consultation with the partners.

**Table 7 children-10-00799-t007:** Parent post-program content feedback.

Feedback Item	# of Responses	Mean/7 (*SD*)	% Response > 5
1. The content of the 1616 Program is compatible with my personal beliefs and values in relation to sport.	82	6.6 (0.7)	100%
2. The 1616 Program is useful.	82	6.2 (1.0)	100%
3. The 1616 Program is credible.	82	6.3 (0.9)	100%
4. The benefits of the 1616 Program are obvious.	82	6.1 (1.2)	100%
5. The 1616 Program is more effective than other youth sports improvement resources.	82	5.3 (1.2)	82%
6. The evidence regarding the impact of being involved with the 1616 Program is available.	82	5.5 (1.2)	91%
7. The 1616 Program was a positive experience.	82	6.4 (0.8)	100%
8. The 1616 Program was well structured and organized.	82	6.2 (0.9)	100%
9. The time requirements for the 1616 Program were appropriate.	82	5.7 (1.3)	91%
10. The delivery of the 1616 Program online was effective.	82	5.9 (1.2)	100%
11. The stories/videos were an effective way to introduce the topics.	82	6.3 (0.8)	100%
12. The reflection items were an effective way to have children think about the topics.	82	6.1 (1.0)	100%
13. The ‘live it outs’ were an effective way to have children engage in behaviors.	82	6.1 (0.9)	100%
14. The 1616 Program helped with positive development for children.	82	6.3 (0.9)	100%
15. I would recommend the 1616 Program to others.	82	6.2 (0.9)	100%

Note. These items were determined in consultation with the partners.

**Table 8 children-10-00799-t008:** Coach post-program content feedback.

Feedback Item	# of Responses	Mean/7 (*SD*)	% Response > 5
1. The content of the 1616 Program is compatible with my personal beliefs and values in relation to sport.	11	6.9 (0.3)	100%
2. The 1616 Program is useful.	11	6.8 (0.4)	100%
3. The 1616 Program is credible.	11	6.8 (0.4)	100%
4. The benefits of the 1616 Program are obvious.	11	6.6 (0.5)	100%
5. The 1616 Program is more effective than other youth sports improvement resources.	11	5.8 (1.1)	82%
6. The evidence regarding the impact of being involved with the 1616 Program is available.	11	6.1 (0.9)	91%
7. The 1616 Program was a positive experience.	11	7 (0.0)	100%
8. The 1616 Program was well structured and organized.	11	6.5 (0.5)	100%
9. The time requirements for the 1616 Program were appropriate.	11	6.3 (0.9)	91%
10. The delivery of the 1616 Program online was effective.	11	6.5 (0.5)	100%
11. The stories/videos were an effective way to introduce the topics.	11	6.5 (0.5)	100%
12. The reflection items were an effective way to have children think about the topics.	11	6.3 (0.8)	100%
13. The ‘live it outs’ were an effective way to have children engage in behaviors.	11	6.5 (0.5)	100%
14. The 1616 Program helped with positive development for children.	11	6.7 (0.5)	100%
15. I would recommend the 1616 Program to others.	11	6.9 (0.3)	100%

Note. These items were determined in consultation with the partners.

## Data Availability

The anonymized data presented in this study are available upon reasonable request from the corresponding author.

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
