# Peer review of "A Proof-of-Concept Evaluation of the 1616 Story-Based Positive Youth Development Program"

_children, 2023, doi:10.3390/children10050799_

Round 1

Reviewer 1 Report

Thank you for the opportunity to review this article. The paper addresses a novel under-researched area, which has the potential to provide useful recommendations for coaches. However, the manuscript is not at the level of "children's" journals and there are important issues that need to be resolved to improve the quality of the manuscript. 

In addition, I have the feeling that I am reading a newspaper article and not a scientific article. There is information that is irrelevant to what is requested in a scientific article. I recommend improving the quality of writing according to the criteria for writing a scientific article.

Specific comments are provided below:

Please, include the affiliation of each author (line 4-5)

INTRODUCTION

Please, remove e.g. and include the reference after de sentence. 

Check the entire document. (line 23)

This section should be included in the methodology section (line 40-47)

Remove i.e. and include 16 through the text (line 43)

Irrelevant information (line 52-55)

Name “IKT” (line 61)

This content should be included in the methods section. Include only the objectives of your study in the introduction section. (112-146)

MATERIAL AND METHODS

Where is the information of “article 2.5”? (line 168)

Include the Code of Ethics Committee (line 169)

Please, include a section of “statistical analysis”

RESULTS

The table description is too long. Please shorten it in tables 1,2,3 and 4.

DISCUSSION

There is no need for subsections in the discussion section, a new thread starts with a paragraph.

Author Response

Thank you for the opportunity to review this article. The paper addresses a novel under-researched area, which has the potential to provide useful recommendations for coaches. However, the manuscript is not at the level of "children's" journals and there are important issues that need to be resolved to improve the quality of the manuscript. 

In addition, I have the feeling that I am reading a newspaper article and not a scientific article. There is information that is irrelevant to what is requested in a scientific article. I recommend improving the quality of writing according to the criteria for writing a scientific article.

RESPONSE: We thank the reviewer for taking the time to review our submission. We appreciate the positive feedback about the potential of our submission and also the helpful suggestions for improving the quality of the manuscript. We have responded to each of the suggestions below.

Specific comments are provided below:

Please, include the affiliation of each author (line 4-5)

RESPONSE: Affiliation for each author appears as superscript on lines 4-5 and written on lines 6-10 per the journal’s guidelines.

INTRODUCTION

Please, remove e.g. and include the reference after de sentence. 

Check the entire document. (line 23)

RESPONSE: Thank you for this suggestion. We have reviewed the document and removed any unnecessary use of, e.g., and have ensured that the reference numbers appear after sentences.

This section should be included in the methodology section (line 40-47)

RESPONSE: We appreciate this suggestion, so have attempted to compromise. Specifically, it is important to introduce the Ladd Foundation and the 1616 program early in the manuscript so that the readers have a good sense of the general context of this article. As such, we have kept the section where it was but have decided to describe the program more thoroughly in the methodology section, as requested.    

Remove i.e. and include 16 through the text (line 43)

RESPONSE: Done as suggested.

Irrelevant information (line 52-55)

RESPONSE: This information was removed.

Name “IKT” (line 61)

RESPONSE: Capital letters were added when “integrated Knowledge Translation” was first introduced to signal the use of the iKT acronym for the remainder of the manuscript. 

This content should be included in the methods section. Include only the objectives of your study in the introduction section. (112-146)

RESPONSE: As suggested, the “PoC Program Overview” section was moved to the Methods section of the manuscript. 

MATERIAL AND METHODS

Where is the information of “article 2.5”? (line 168)

RESPONSE: After review, we did not require ‘article 2.5’, so it has now been removed from the text.

Include the Code of Ethics Committee (line 169)

RESPONSE: As requested by the editorial board, the statement of institutional ethics approval for this project is provided on lines 652-654. 

Please, include a section of “statistical analysis”

RESPONSE: We included a general section of evaluation design and analysis on lines 231-251. Several analyses were conducted and explained when describing how we addressed each of our research objectives. 

RESULTS

The table description is too long. Please shorten it in tables 1,2,3 and 4.

RESPONSE: We have reviewed each Table description. Importantly, Tables must be able to stand alone. So we feel that the information in the Table descriptions is comprehensive, relevant, and essential for a complete understanding of our data.    

DISCUSSION

There is no need for subsections in the discussion section, a new thread starts with a paragraph.

RESPONSE: We appreciate this comment, but we believe that the two headings of the Discussion, which focus on the two objectives of our study, provide a clear delineation of the outcome and process evaluation we conducted and avoid confusion in light of a large amount of data presented.

Reviewer 2 Report

I have carefully read the manuscript and my opinion is that the manuscript has a merit to be published in your reputable journal with some minor corrections. The manuscript is original, informative and readable. The authors tried to evaluate whether the 1616 Program ‘worked’ in enhancing PYD-related outcomes in youth, as well as to determine if the concepts were engaging and enjoyable for youth, their parents, and coaches. Judging from the sports scientist point of view, the structure of the abstract should be in the following order: the purpose of the study, method, results and short discussion with conclusions, so I would appreciate if the authors apply it as well as extend the abstract method, results and conclusion and offer potential readers to get a bit more information. On the other hand, the introduction is well written but I would appreciate if the authors include purpose of the study as a part of the introduction, while the PoC Program Overview should be a part of the method section. The method section should divided into three section: population, variables and statistical procedures, as well as more precisely describe the three issues. I would also appreciate if Results and Discussion sections do not contains subsections. I would recommend to the authors to prepare the conclusion part in the following order: the main conclusions, the limitations of the study (more precisely) as well as recommendations for the further studies (it is very important to briefly elaborate it and highlight the most important notes, mostly due to the fact that this manuscript has an excellent practical application). 

Author Response

I have carefully read the manuscript and my opinion is that the manuscript has a merit to be published in your reputable journal with some minor corrections. The manuscript is original, informative and readable. The authors tried to evaluate whether the 1616 Program ‘worked’ in enhancing PYD-related outcomes in youth, as well as to determine if the concepts were engaging and enjoyable for youth, their parents, and coaches.

RESPONSE: Thank you!

Judging from the sports scientist point of view, the structure of the abstract should be in the following order: the purpose of the study, method, results and short discussion with conclusions, so I would appreciate if the authors apply it as well as extend the abstract method, results and conclusion and offer potential readers to get a bit more information.

RESPONSE: The abstract was rewritten to integrate these suggestions.   

On the other hand, the introduction is well written but I would appreciate if the authors include purpose of the study as a part of the introduction, while the PoC Program Overview should be a part of the method section.

RESPONSE: As suggested, the PoC program overview was moved to the methods section. The purpose of the study now appears in the last paragraph of the introduction. 

The method section should divided into three section: population, variables and statistical procedures, as well as more precisely describe the three issues.

RESPONSE: Although we appreciate this suggestion, the method section addresses the three issues listed in this comment. The headings and content were adapted to the specificity of the qualitative and quantitative data we presented and so a revision would do a disservice to the readability of the manuscript.    

I would also appreciate if Results and Discussion sections do not contains subsections. I would recommend to the authors to prepare the conclusion part in the following order: the main conclusions, the limitations of the study (more precisely) as well as recommendations for the further studies (it is very important to briefly elaborate it and highlight the most important notes, mostly due to the fact that this manuscript has an excellent practical application). 

RESPONSE: As requested, we tried to limit the subsections in the Results and Discussion. Because of the mixed method (qualitative and quantitative), we presented the conclusions, limitations, and recommendations throughout the results and discussion sections.   

Reviewer 3 Report

Congratulations. The subject is very interesting.

Round 2

Reviewer 1 Report

No creo que el manuscrito revisado haya realizado las modificaciones sustanciales para abordar mis preocupaciones. Por lo tanto, sigo pensando que este manuscrito no es apto para ser aceptado.

Reviewer 2 Report

I have no further requests.